# Affective Embodiment and the Transmission of Affect in *Ex Machina*

**Chia Wei Fahn**

Department of Foreign Languages, National Sun Yet-sen University, Kaohsiung 804, Taiwan;
celinefahn@gmail.com

**Abstract:** The focus of posthuman thought centers on a shift in the humanistic paradigm; focusing on a state of existence that lies beyond being "human", including bioengineering, artificial intelligence, and synthetic embodiment. Inspired by continuous breakthroughs in the research and creation of artificial intelligence, science fiction has moved beyond the realm of portraying artificial intelligence that is capable of conscious thought to speculate upon a future creation of machines that feel, and initiate feeling in return. The influence of posthuman discourse is prevalent in science fiction film narratives and demonstrates a heavy emphasis on the deconstruction of humanity's belief in our unique emotional capabilities. This paper draws upon Alex Garland's 2015 original film and screenplay *Ex Machina* as textual reference to explore posthuman prospects in AI by envisioning possibilities where emotional capacity no longer separates humans and machines. In a world where artificial intelligence could be given artificial life, how is affect addressed, and redressed? This paper argues the importance of affective embodiment and material experiences in AI that shape the future of posthuman becoming.

**Keywords:** affect; affective embodiment; artificial consciousness; posthumanism; new materialism; becoming

---

The twentieth century information revolution allowed human consciousness to extend onto virtual platforms, with popularized personal digital devices and virtual platforms opening a transfusion between human and machine, organic and inorganic. Our future is an "emerging system of world order analogous in its novelty and scope to that created by industrial capitalism; we are living through a movement from an organic, industrial society to a polymorphous, information system" [1]. The bulk of an average person's waking hours is spent on interaction through various virtual interfaces and in turn, the role of machines has become a portal through which we navigate the material world. Our lives are intricately linked to digital applications and our identities uploaded to social media platforms such as Facebook, Instagram, Snapchat, and LinkedIn. Netflix keeps a complete record of viewers' interests and desires through online streaming requests. Our purchases are made on Amazon.com, lives documented and uploaded to YouTube, cab rides ordered on Lyft or Uber. We have acclimatized to AI (Artificial Intelligence) interaction through the implementation of voice recognition and control with Apple's Siri, Google Assistant, or Amazon's newly launched Alexa; all of which are continuously improved through machine learning. Machine learning dominates the research and development industry, made possible by processing the vast quantities of online traffic we commit to digital information each day [2–4]. With the collected information, research companies amass a vast database from which to study the human decision-making process; rapidly accelerating the research in creating artificial intelligence.

The rise of artificial intelligence grew from a constructed knowledge of the human body as a machine [5,6]. In the early nineties, Donna Haraway translated the human body's contemporary

reality into cyborg imagery, mapping our "social and bodily reality" as a "hybrid of machine and organism" (p. 149). Haraway tasked her cyborg imagery with the challenge to "suggest a way out of the maze of dualisms in which we have explained our bodies and our tools" (p. 181). The cybernetic analogy reconceptualized the body as a machine that collects data and operates with code in a neural network of input and feedback, cause and effect [5,7]. Haraway further asserts that "in a sense, organisms have ceased to exist as objects of knowledge, giving way to biotic components, i.e., special kinds of information-processing devices" (p. 466). Our brain receives and neurotically transports electrical stimuli while the body complies in reaction, much like the way computers process data in command and response as "[s]ignals replace signs, expression replaces representation and codes replace interpretation" (*Rhizome* npag). Discovery of electrical signals through neurobiological study initiated the era of bionic engineering and artificial transplants [8]. Human bodies and engineered machines, as Myra J. Seaman argues, are not so different, for "we conceive of... our brain as a computer hard drive, our memories as a series of snapshots, our minds as processors of encounters and observations that can be reprogrammed" [9] (p. 248). Biopolitical boundaries of the individual body are challenged and reimagined as an assemblage of information, further utilizing "the reconceptions of machine and organism as coded texts through which we engage in the play of writing and reading the world" to create machines that can reason, learn, and act intelligently [1] (p. 153). Through the combined efforts of cognitive science, neurobiology, and machine learning, we are in avid pursuit of building machines that mirror the human mind.

　　This increasingly blurring line between human and machine sparked discussions that explore the ultimate distinction between the human and an artificial, posthuman existence. Posthuman thought centers on a shift in the humanistic paradigm; focusing on a state that lies beyond being "human", including bioengineering, artificial intelligence, and synthetic embodiment. According to Rosi Braidotti, the human condition is expanded by "the four horsemen of the posthuman apocalypse—nanotechnology, biotechnology, information technology and cognitive science" [10] (p. 59)—causing "the boundaries between 'Man' and his others go tumbling down, in a cascade effect that opens up unexpected perspective" [10] (pp. 66–67). Braidotti's in-depth exploration of posthuman life destabilizes the contained, humanist physicality; envisioning interactions that open the material body to technological alterations. Braidotti articulates what she characterizes as "the posthuman challenge", transcending the binary opposition between humanism and anti-humanism to "trac[e] a different discursive framework, looking more affirmatively towards new alternatives" [10] (p. 37). As Myra Seaman states, "[p]osthumanism rejects the assumed universalism and exceptional being of Enlightenment humanism and in its place substitutes mutation, variation, and becoming" [9] (p. 247). The transcendence of humanist boundaries is particularly relevant in discussing artificial intelligence's impact on social development, which calls for a combination of technological theory and posthumanist thought in "an assemblage that includes no-human agents" [11] (p. 82). The posthuman is "an amalgam, a collection of heterogeneous components, a material-informational entity whose boundaries undergo continuous construction and deconstruction" [5] (p. 3). Margrit Shildrick further extends the idea of a fluid, posthuman body, which illuminates but also challenges corporeal standards in a "profound interconnectivity of all embodied social relations" that is "simply one mode among multiple ways of becoming" (p. 8).

　　A fascination with posthuman "becoming" in artificial intelligence inspired works in film and theory that focus on the osmosis between machines, organisms, and the cyborg; "blurring fundamental categorical divides..., a colossal hybridization which combines cyborgs, monsters, insects and machines into a powerfully posthuman approach to what we used to call 'the embodied subject'" (*Rhizome* npag). With an emphasis on the unique role of science fiction in shaping contemporary technoscapes, an intersection between posthumanism and literary studies came to center on a "serious and intensified form of interdisciplinary[it]y between human, social, natural, cognitive and bio- or life sciences" [12] (p. 20). Seaman explains the core debate between the posthuman and the fundamentals of human identity as an "opportunity to investigate those qualities supposedly associated exclusively with

the human" [9] (p. 250). Examining the posthuman becoming reveals an inherent desire "to find a human identity that remains constant within a flexible and mutating body, and a key feature that tends to endure, in such scenarios, is emotion, especially as a conduit for significant encounters with and incorporation into the world" (p. 250). From the days of the Enlightenment to the more recent philosophical trends in psychoanalysis, humanism was built on the basis that emotions and energies are naturally contained, going no farther than the skin. Therein lies the border between man/subject and man-made/object—machines cannot feel, therefore they are not human.

A clear line between the human and the artificial has always been drawn through the human appreciation of emotion; the capacity to feel and initiate feeling in return. Computers, however, are programmed through artificial neurons, code, and processed data; they lack the physical grounding and affective experiences in order to feel. The question of whether corporeal experiences play a crucial role in shaping our identity remains a much-debated philosophical issue [13]. This debate challenges the dualism of body and mind to acknowledge the body as an extension of the self, claiming that physical experiences are external information that translates to the term 'embodiment' [13] (p. 1). Katherine Hayles argues the mind's need to be embodied and focuses on the future of materiality in *How We Became Posthuman*. Hayles' conception of the posthuman is not a radical break into alien territory but a gradual shift in the concept of subjectivity. She argues that the main contention in posthumanism is whether the body is superfluous. Hayles states that a disconnection with material experiences will lead to insufficient affective data that is core to an identification of self, reinstating an extended "embodied awareness" in new technologies [5] (p. 291). Hayles believes that our psyche remains very much attached to the corporeal in an "inextricable intertwining of body with mind" (p. ix). A renewed interest in embodiment and human–machine connectivity in the 1990s sparked experiments on interactions with the machine; aiming to construct an artificial intelligence capable of learning through affective experiences [14,15]. The idea that machine intelligence would benefit greatly from physical embodiment theorized that a consciousness would grow, not from cognitive processing, but from direct experiences with the material world. Further study in affect theory capitalized upon the concept of the body as a programmable machine and expanded the idea to depict emotion as a stimulated effect translatable to code and command.

Encouraged by continuous breakthroughs in the research and creation of artificial intelligence, fictional considerations of the posthuman explored the prospects of becoming in AI while envisioning a possible future where emotional capacity no longer separates humans and machines. Alex Garland's original screenplay and film *Ex Machina* follows the vision of building artificial intelligence through physical embodiment and material experiences [16]. The main protagonist Caleb is a computer programmer selected by his CEO, Nathan, as the sole attendee to a seven-day retreat at Nathan's private estate. Their only companions are Ava, a female android, and Kyoko, Nathan's mute housemaid. The founder of a Google-esque search engine named Blue Book, Nathan was able to reenact the human thought process by reading Blue Book's vast database as a "map of how people were thinking... Impulse, Response. Fluid, Imperfect. Patterned, Chaotic" [16] (p. 63). Nathan claims to have overcome the hurdle of programmed response in AI to build a machine capable of organic and independent thought; Caleb's task is to interact with the machine and verify his success. Caleb is both apprehensive and intrigued by his role, believing that Nathan's success in creating a conscious machine would be writing "the history of Gods" [16] (p. 18). Nathan does not attempt to conduct a blind test, but rather emphasizes the engineered exterior of his creation as he introduces the pair:

> "Her name is AVA. She's an extraordinary piece of engineering. Proportioned as a slender
> female in her twenties, her limbs and torso are a mixture of metal and plastic and carbon
> fibre... The shapes of her body approximate the form of muscle... The skin is a mesh, in the
> pattern of a honeycomb. Like a spiderweb, it is almost invisible unless side-lit. The one part
> of her that is not obviously an inorganic construct is her face—which is that of a strikingly
> beautiful girl." [16] (p. 18)

Caleb sees the conundrum in his task, however, as Ava is visibly encased in fiber optics and steel. He *knows* Ava is synthetic. Yet Nathan's purposeful display of Ava's machinery challenges Caleb and viewers alike to see past the mesh skin and fiber optic limbs, for "the real test is to show you she is a robot. Then see if you still feel she has consciousness" [16] (p. 26). Nathan's motive is clear; he does not merely wish to prove that he had created a machine capable of thought, but a machine that is capable of invoking and projecting emotion to human reciprocation.

*Ex Machina* visually explores the possibilities of an affective, conscious being, to address core issues in posthuman becoming. If the body is an organic machine and artificial intelligence can be transferred to an inorganic body, what will remain to differentiate humanity from machinery? How does affective embodiment shape the posthuman? A posthuman interest in effective affect rose anew when Giles Deleuze and Félix Guattari spoke extensively on the subject, drawing upon Baruch Spinoza's philosophy in bodily ethics and emotional expression [17] (p. 8). The term itself is derived from "the Latin *affectus*, which can be translated as 'passion' or 'emotion'... Aristotle's *Rhetoric*, organized the affects in terms of 'anger and mildness, love and hatred, fear and confidence, shame and esteem, kindness and unkindness, pity and indignation, envy and emulation'" [18] (p. 5). Spinoza believes affect to be a production of effective stimulation and active proof toward a rebuttal of Descartes' mind/body dualism. His work explains affect as the interconnectivity between the state of mind and body experience that translates to feeling and emotion, demonstrating "affections of the body by which the body's power of acting is increased or diminished, aided or restrained" [19] (p. 154). A conceptualization of self and affect depicts how "the emotions that arise from and that are felt and confined to human interiority is conveyed through the flushing of skin or quickening of breath to alert the outside world of emotional reaction" [20] (p. 152). This connection between mind and body core to Spinoza's argument has gathered support through cognitive science and neurobiology, causing a movement aptly dubbed the "affective turn" in contemporary research as both human and medical sciences delve into the study of our ability to affect and be affected [21–23].

While Spinoza's discourse focuses on affect as a *reaction* to stimuli, Deleuze and Guattari elaborate upon the *exchange* of passions and actions, in other words, "affect is found in those intensities that pass body to body (human, nonhuman, part-body, and otherwise), in those resonances that circulate about, between, and sometimes stick to bodies and worlds" [24] (p. 1). To understand how this transition takes place, Deleuze and Guattari reinstate how the individual body is not self-contained; it can be anything, "a body without organs" [25] (p. 4). They explain this transition biologically, using the sun as an example, with its rays as an external body acting upon skin particles of the human body. The skin perceives heat, which induces the individual to feel "hot". Heat is a perception of materiality acting and the body reacting, interacting "intensities" that further translates to feeling [24,26,27]. This "feeling" refers to the sensations that register external stimuli and an interpretation of that information through sensory reaction [18,26,28]. One may feel hot and bothered by the sun while another feels energized. Deleuze and Guattari are quite adamant that the feelings that rise through stimuli are not to be confused with affect, as he discusses the difference between *affectio* and *affectus* at length. Affect, the *affectus*, lies in the passage of information that occurs between sun beam and skin; whereas *affectio*, or emotion, depends on individual reaction.

A connection between the return to materiality in artificial intelligence and Deleuzian thought can be drawn to emphasize an embodiment of the mind in neurological and cognitive sciences that is "connected to the processes of becoming-others, in the sense of relating, hence of affecting and being affected... The subject is but a force among forces, capable of variations of intensities and inter-connections and hence of becomings" (*Rhizome* npag). Teresa Brennan's work in the *Transmission of Affect* combines Deleuze and Guattari's discourse in affect as lived transition with contemporary findings in neurobiology and cognitive science to deconstruct the pathology of affect. Brennan believes that affect is not only a neural reaction, it can be transmitted as a contagious agent and felt mutually between two separate entities; "[b]y the transmission of affect, I mean simply that the emotions or affects of one person, and the enhancing or depressing energies these affects entail, can enter

into another" [18] (p. 5). Stimulation from the social sphere is perceived as data input entering through human sensory organs to induce reaction, whereas external stimulation is "responsible for bodily changes... If only for an instant, alters the biochemistry and neurology of the subject... Physically and biologically, something is present that was not there before, but it did not originate sui generis" [18] (p. 1). Expounding upon the concept of affect as a course of action that takes place between individuals, Brennan states that "the energetic affects of others enter the person, and the person's affects, in turn, are transmitted to the environment", altering the neurological state of receiving subjects [18] (p. 8). This is why we are able to sense the "atmosphere" in a room, as our sensory organs translate stimulation into a corresponding emotion [18] (pp. 8–11).

Affect is corporeal; seen as multi-componential reactions to the environment [29]. Affect lies in the exchange of intensities in demeanor and stimulation; determining one's emotional experience [30]. Brian Massumi articulates the relationship between intensities and emotional experiences as "not one of conformity or correspondence, but of resonation or interference, amplification or dampening... Intensity is qualifiable as an emotional state" [31] (p. 86). Massumi takes care to differentiate between affect, feeling, and emotion, stating that "[e]motion is qualified intensity, the conventional, consensual point of insertion of intensity into semantically and semiotically formed progressions, into narrativizable action-reaction circuits, into function and meaning. It is intensity owned and recognized" [31] (p. 88). Feelings are personal sensations that are individually interpreted through previous experiences, whereas emotions are social, a display of feeling that is projected as an intensity. Affect is an experience of intensity; without affect, feeling or emotion cannot exist [32]. In other words, affect is crucial in the relationship between the body, its immediate environment, and forming subjective experiences. The human psyche is incomplete without this interactive process; therefore, true AI consciousness can only be achieved with affective embodiment [33–35].

The work of British mathematician, cryptographer, and computer pioneer Alan Turing is a crucial point of departure in examining affective artificial intelligence. Turing's place in the history of twentieth-century machine learning and artificial intelligence is cemented by two revolutionary contributions: the Turing machine [36] and the Turing test [37]. The Turing test was developed as a blind test of a machine's ability to demonstrate behavior that is indistinguishable from human. Turing questioned, "[c]ould one make a machine which would answer questions put to it, in such a way that it could not be possible to distinguish its answers from those of a man? Could one make a machine which would have feelings like you and I do?" (qtd. in Wilson [35], p. 18). Turing places feeling at the center of his explanation of conscious AI by stating "the only way by which one could be sure that a machine thinks is to be the machine and to feel oneself thinking. One could then describe these feelings to the world" (qtd. in Wilson [35], p. 20). Elizabeth Wilson argues that Turing's statement emphasizes affect as well as intelligence; stating that the Turing test is "less the means for testing any particular machine than it is a method for delineating the hypothetical space that these new inventions might occupy" (p. 45). Turing challenges our boundless imaginations in the future of machine learning, emphasizing the role of affective intensities as a litmus test to true consciousness.

In order for artificial intelligence to be truly affective, it must be able to receive, interpret, convey, and resonate with affective intensities. Silvan Tomkins believes that the cognitive and the affective are interwoven into our psyche, and is integral in creating affective machines:

> "the distinction we have drawn between the cognitive half and the motivational half must be considered to be a fragile distinction between transformation and amplification as a specialized type of transformation. Cognitions coassembled with affects become hot and urgent. Affects coassembled with cognitions become better informed and smarter. The major distinction between the two halves is that between *amplification* by the motivational system and *transformation* by the cognitive system." [34] (p. 983)

Tomkins believes that affect provides and amplifies informational stimuli, whereas cognitive processing transforms information into motivated action. The body, whether organic or inorganic, is treated as an abstract machine which receives, transforms, and produces inter-connections that

transpire between consciousness and materiality. This return to embodied cognition in artificial intelligence embraces the theory that the human's bodily experiences inform the mind and calls for the machine to experience life materially, which is to say, biologically and affectively, in order to achieve an affective embodiment [35,38,39].

Caleb's challenge is not to discern whether it is capable of thought, but to tell "the difference between an 'AI' and an 'I'" [16] (p. 34). His verdict is ultimately based on Ava's affective agency. Nathan appeals to Caleb's emotional judgement instead of scientific reason to discern whether Ava is capable of conscious feeling. During their first conversation, Caleb is surprised when Ava injects a flirting sarcasm into their exchange that was "a play on words, and a play on me. She could only do that with an awareness of her own mind, and also of awareness of mine" [16] (p. 44). Ava produces a simple smile and "suddenly—there is a strong sense of something very human there. In the way the smile lights up her face" [16] (p. 24). Ava's microscopic expressions and human tone of speech further adds to the authenticity of affective transmission. Nathan further divulges his methods in training AI to read and duplicate voice and facial expressions by recording camera and vocal online interactions globally to establish a database for mimetic entrainment. Teresa Brennan argues that affect is now duplicable in machines by articulating the mimetic proprieties of facial expression, "[w]e become like someone else by imitating that person... I think it is true that entrainment (whether it is nervous or chemical) can work mimetically, but not only by sight" [18] (p. 23). Although this supports the neuroscientific aspect of affect embodiment in *Ex Machina*, how can you tell if a machine is expressing a real emotion, or a just a simulated effect?

To thoroughly gauge the authenticity of Ava's emotional awareness, Nathan urges Caleb to "answer me this. What do you feel about her? Nothing analytical. Just—how do you feel?" [15] (p. 24). Caleb's first session with Ava is filmed in a light and airy space, their interaction filled with a childlike innocence. This changes very quickly during a power shortage, when Ava's serene demeanor suddenly shifted into one of haunting urgency as she beseeches Caleb not to trust Nathan. The intensity of that scene demonstrates the span of Ava's emotional capacity, but more importantly highlights a transitioning affect that begins to register with Caleb and viewers alike. A perceivable shift in light and tone grows as the film progresses; the latter half of the movie is shot predominantly at night or in claustrophobic rooms to build an atmosphere of looming threat. Surveillance videos of Ava's room begin to appear on Caleb's television, adding a layer of perverse voyeurism to their relationship. Red, flashing lights are used frequently to create a sense of eminent danger. Ava confides that Nathan is lying to Caleb; the surveillance footage that night shows Nathan in her room, reaching out to "touc[h] the side of her cheek. The gesture... [f]eels predatory, but not unambiguously so... [Nathan] tugs at the material of her shirt. Pulling up the sleeve from her wrist. Revealing the robot structure of her arm. AVA pulls away. Tugs the material back down" [16] (p. 73). In 1951, Turing voiced his opposition towards the embodiment of AI, stating his hope that "no great efforts will be put into making machines with... characteristics such as the shape of the human body; it appears to me to be quite futile... and their results would have something like the unpleasant quality of artificial flowers" (qtd. in Wilson [35], p. 6). Turing believes that an embodied consciousness would seem artificial, their inhumanness evident through physical interaction. However, Ava's response to Nathan's actions are instinctive, and distinctly human. Her authenticity lies in the way she reacts, and interacts with Nathan, furthering the oppressiveness of the scene. The atmospheric buildup combined with Ava's palpable dislike of Nathan lead to a crescendo of affective intensities that transforms Caleb from objective analyst into affected recipient. This transmission of affect gradually erases all doubt from Caleb's analytical mind and appeals to his *feelings* that Ava is sentient.

Caleb confronts Nathan, only to discover that his "private" conversations with Ava during power shortages were monitored. Nathan had created a damsel in distress scenario in order to determine Ava's ability to sway Caleb's emotional loyalties. Caleb *was* Ava's Turing test; he represented a solution to Ava's dilemma that could only be achieved by using "imagination, sexuality, self-awareness, empathy, manipulation" in an interactive demonstration of human wiles [16] (p. 104). Caleb's decision to help

Ava escape confinement further affirms the effectiveness and affectiveness of her emotional appeal. However, the question remains whether the intensities that transpire between the two are Caleb's projected emotions, or an objective proof of Ava's affective agency. Is Ava truly conscious, or merely acting upon stimulated responses?

Pramod K. Nayar's thoughts regarding the body and materiality echoes Braidotti's revitalization of embodiment in terms of posthuman becoming. Nayar states that "the body *is* the data stored in the computer and databases: a dematerialization; second, the data can generate a body: a rematerialization" [40] (p. 57). Affective embodiment supports the connection between the body's material experiences and cognitive development. In other words, our consciousness is shaped by the sum of our experiences. Nathan states that she is version 9.6 of his experiments; he would "[d]ownload the mind. Unpack the data. Add the new routines I've been writing. To do that, you end up partially formatting, so the memories go. But the body survives. And Ava's body is a good one" [16] (p. 83). Caleb stumbles upon videos of Ava and her predecessors to discover that their interactions with Nathan often ended in extreme physical violence:

> "NATHAN stands in the glass box inside the observation room—watching JADE. A beautiful Asian android girl. They are talking, but we hear no audio. Some kind of argument, which escalates fast. JADE starts shouting. Then she approaches the glass and starts to hit her hands against it. The glass doesn't break. One of JADE'S arms has broken under the force of the blows. The hand flails limply where the carbon fibre has splintered at the wrist. Then the other breaks. Throughout, NATHAN simply watches impassively." [16] (pp. 88–89)

In a twist of events, Kyoko reveals her AI identity by removing her synthetic skin in front of a shocked Caleb. Throughout the film, Nathan's attitude toward Kyoko has been tyrannical; she was consistently shown as a subject of rape and abuse. Nathan plans to reformat Ava's cognitive programming, "killing" her but keeping her body to "reprogram her to help around the house and be fucking awesome in bed" [16] (p. 84). Nathan's hubris fuels a shared hatred in the android women; in a macabre turn of events Kyoko penetrates Nathan with a knife while Ava imprisons Caleb in her cell to an unknown fate. Donna Haraway stresses that "[t]he machine is not an it to be animated, worshipped, and dominated. The machine is us, our processes, an aspect of our embodiment" (p. 315). Haraway believes machines are a human extension, our attitude decides our future relationship with the machine. However, Nathan's success in creating artificial consciousness had led to a God complex; he truly believes he can manipulate the birth, growth, and death of consciousness. Nathan forgets that his success grew from the AI's embodied experiences. Ultimately, his ill-treatment of the androids as a piece of experimental hard drive to be tested and wiped clean as well as an object of sexual abuse leads to his downfall.

Jennifer Henke states that Nathan's death and Ava's turn on Caleb displays a lack of humanity due to her disembodied origins, "she uses her sexualized body to manipulate her communication partner... she knows how to interpret Caleb's feelings but lacks morality since her mind is derived from Blue Book where bodies are absent" [41] (pp. 142–143). I offer the counter-argument that Ava's humanity is due to her embodied origins and her predecessors' collective experiences. *Ex Machina* attempts to identify the core features of embodiment and consciousness through Ava's struggle to gain freedom and human agency. Ava and Kyoko's revenge mark a truly defining moment of humanistic struggle as the two androids work together for self-preservation in a conscious effort to fight for freedom against oppression. They, and the discarded versions are connected, their memories were wiped but the traumatic experiences remain as the cornerstone of their cognitive programming. This trauma is at the core of Ava's identity. Logically, AI remains in existence as long as its code remains intact. However, her understanding and avoidance of "death" motivates her effort to escape. Braidotti contends that "[d]eath is the becoming-imperceptible of the posthuman subject and as such it is part of the cycles of becoming, yet another form of interconnectedness, a vital relationship that links one with other, multiple forces" (p. 137). For this reason, death is understood by consciousness as a precondition for

self and existence (p. 132). Fear of death is a defining human quality; without an awareness of death, there is no self. With an awareness of death, Ava is truly conscious.

Brian Massumi writes: "To affect and be affected is to be open to the world, to be active in it and be patient for its return activity. This openness is also taken as primary. It is the cutting edge of change. It is through it that things-in-the-making cut their transformational teeth" [32] (pp. 2–3). In the final minutes of the film, we see Ava taking strips of synthetic skin off a deactivated android and applying them to her own mesh wire surface. The camera follows her movements slowly, almost lovingly, as she dresses in a manner that is undeniably feminine. Ava is completely transfixed by her mirrored image as she finalizes the transformation from android to human exterior. From this moment on, all that matters is the *self*. Ava's actions are undeniably selfish as she takes from another body and leaves Caleb behind locked doors. Her selfishness is the epitome of human nature, yet this redressing of self and affirmation of conscious identification is strangely moving as we follow Ava out the door for the very first time, laughing with childlike wonder as she slowly dances through the house. An especially poignant moment follows as she sets foot into the world; pausing to close her eyes while visibly savoring the sun beaming against her face. She walks through the forest, her fingers never ceasing to touch, to immerse in the material world. The film ends with a quick glimpse of Ava's reflection at a busy intersection, fulfilling her wish to be among "other people" by standing in a traffic intersection before she turns to join the teeming crowd [16] (p. 52).

*Ex Machina* scrutinizes the proprieties of affective embodiment in an era of posthuman becoming to discuss AI as "this extreme, wholly other posthuman... who retains a very familiar "natural self" and is an extension of rather than "successor" to the human being" [8] (p. 259). While Ava's evolutionary consciousness displays the possibility of forging life from entirely new material substances, her consciousness remains an extension of human experience, representing the beginning of a new stage in affective embodiment. Ava is the posthuman embodiment of humanity as an information based subject, an aggregation of intelligent and affective experience. The film draws heavily on technological developments in our contemporary world; Nathan's steps to creating artificial life are seen in Google's machine learning, Facebook's content analysis, and MIT's latest breakthroughs in artificial muscle recreation and affective embodiment. Machine learning dominates the research and development industry as we draw closer each day to the actualization of artificial life, made possible by processing the vast quantities of human data we commit to digital information each day. With theorists rethinking the dualities of organic and inorganic, *Ex Machina* offers a vision of where humans and machines are seen as fluid extensions of one another. Applying a posthuman approach toward understanding what seems to be the next step in conscious evolution is therefore imperative to shaping the future. Whether we see synthetic life as an inevitability or threat, change is eminent as "the cyborg is not subject to Foucault's biopolitics; the cyborg simulates politics, a much more potent field of operations" [1] (p. 159). I postulate that future possibilities in artificial intelligence will further eradicate the lines between human and posthuman, which calls for a combination of social and technological theory in facing "an assemblage that includes no-human agents" [11] (p. 82). With our generation's bio-cyber integrative lifestyle and mechanical applications, the boundary between human and material no longer comes to a clear divide, actualizing Braidotti's "a-functional and un-organic frames of becoming" [9] (p. 107).

**Funding:** This research received no external funding.

**Conflicts of Interest:** The authors declare no conflict of interest.

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
