# Peer review of "Affective Embodiment and the Transmission of Affect in Ex Machina"

_philosophies, doi:10.3390/philosophies4030053_

Round 1

Reviewer 1 Report

This paper raises good points about the consideration of affective embodiments in examinations of cyborg narratives but requires substantial edits tosharpen the argument. The author needs to be much more clear in the presentation of their central research statement or argument. By page 5, readers are still in the dark about where this paper is heading or what its central argument is. 

The move to an analysis of Ex Machina needs to be prepared early on and carefully argued. At times the connection between the first theoretical half of the paper and the thick description of Ex Machina remains notional. Deleuzian scholarship on affect is not picked up and also new materialist scholarship is barely mobilized for a critical analysis of the film. The first parts reads too much like an overview or summary and is lacking a clear critical argumentative structure. I suggest the author carefully reviews this text, interweaving both parts and foregrounding a clear argument and through-line from the beginning. (this needs to include the abstract, which does not include a research statement or argument, and also does not mention Ex Machina and only vaguely refers to “film narrative”). The fields of transhumanism and posthumanism are nowhere clearly defined (despite their important differences). See for instance Raulerson’s book Singularities: Technoculture, Transhumanism, and Science Fiction, or the recent Cambridge Companion to Literature and the Posthuman. The author’s consideration of affect theory remain at times vague and would benefit from references to the work of e.g. Brian Massumi or Lisa Blackman. 

The author also needs to review for idiomatic language use. While the diction is generally sophisticated and appropriate, some passages remain unclear, because of awkward phrasing and grammatical errors. 

I suggest the authors does a search/replace for all references to “man” and replace it with “human.”

the transition to film analysis is too abrupt and requires cues as to where the analysis of the film is going. 

the bibliographical reference of the film script is not clear. 

More detailed copy-edits below.

—————

7

abstract should not include quotes. 

20

the controversial aspect of Brennan’s views is not taken up below. 

also consider adding Guattari. 

29 establishing

35 incongruent plural/singular: body-machines

39-40

double-check grammar:

 boundaries between our bodies and the material world are made “permeable between tool and myth, instrument and concept, historical systems of social relations 40 and historical anatomies of possible bodies” (162). 

44

unclear bibliographical reference

88

unclear: “interaction between each gene, the temporal sequence of external environments, and physical change”

-> what is the termporal sequence of environments? how does it differ from physical change? How do genes “interact” with physical change or temporal sequences?

114-115

“Our generation spends more time interacting on a virtual interface as opposed to maintaining 114 human connection”

-> this point is a little clichéd and relies on an essentialist idea of both the human and connections.

132

“These questions give rise to new material philosophy”

-> new materialist philosophy?

145

“study 144 in individual humanism”

-> what is individual humanism?

“that emotions and energies are naturally contained“ 

-> unclear. Is energies meant metaphorically here? for what? if not, this seems like an untenable point.  

146

“man/subject and man-made/object” 

-> unless critically reflected this perpetuation of the male-gendered human is problematic.

148

“A clear line between the human and the artificial has always been drawn through the appreciation of emotion.”

-> review use of “always” — unnecessary generalization. 

151

“definitive studies in Geno-determinism”

-> what are definitive studies? 

156

“is derived “the Latin affectus “

-> derived FROM

164

“depicts”

-> consider more suitable word

172

(Webdeluze 172 n.p.) 

-> typo

-> link in bibliography does not work.

-> is this a translation by the author?

-> missing sentence marker after parenthesis

187

As man relies increasingly 

-> inappropriate default gender.

189

-> “the” part of title of Brennan’s book?

191

“the pathology of affect”

-> what is this? why definite article? 

-> To speak of pathologies in this regard, seems like a subjective judgment/evaluation

209

“Posthumanists believe that “[n]o objects, spaces, or bodies are sacred in themselves; any component can be interfaced with any other if the proper standard, the proper code, can be constructed for processing signals in a common language” (Haraway 161).” 

-> this is an uncritical generalization of “posthumanists”. Undoubtedly there are many posthumanisms and many different strands within the field, many of which reject metaphors of codes and signals”

212-214

“we now understand that from sex and reproduction to growth or aggression: all fields in human action involve the trigger of hormonal messages.”

-> consider rephrasing: fields in human action (unidiomatic)

-> triggering

-> unclear meaning of sex. Sexual activity? physiological sex?

218-221

“The actuality of artificial life consciously manifesting emotions grows in possibility, urging modern film to gradually depart from the popular dystopian formula of apocalyptic inducing machine overlords and began to explore the possibilities of a feeling, inorganic existence in artificial intelligence.”

-> this sentence is ungrammatical. Please review. 

-> this move to film is very abrupt (requires more context) and the argument a little to teleological and vague. (“urging modern film”)

221-223

“Humanity is no longer infinite: with the era of posthuman constantly being redefined, the subject of a transcendent entity and the emotional passage which flow between man and machine discussed through an examination of affect in film narrative.”

grammatically unclear 

point incomprehensible

when was humanity infinite? what does this mean?

inadequate use of “man”

“with the era of posthuman”… (requires article)

228

“budgeted film”

-> unclear

238-242

-> this reference is unclear. (biblio-reference in parenthesis needs to be adjusted)

-> if this a reference to the written screenplay, then the reference in the bibliography is also unclear. Please provide a link or publication details. 

244

“The AI is undoubtedly housed in a synthetic body”

-> unidiomatic use of “undoubtedly”

257

“humanistic” 

-> review. find different word. 

271

sees

325

“and humanity ceases to be without emotion.”

-> unclear use of “cease”. has it ever been without emotion?

364

“Applying a posthuman approach toward understanding what seems to be the next step in conscious evolution is therefore imperative to shaping the future.” 

-> unclear and vague reference to “posthuman approach.” What is a posthuman approach? an approach that articulates a fervent critique of humanism and anthropocentrism? 

378-380

A conscious shift in Cyborg fiction began with Roy Battey’s famous “tears 378 in the rain” monologue in Blade Runner, progressing with Robin William’s fight for legal 379 acknowledgement of his humanity in Bicentennial Man and focusing on the discussion of affect 380 transmission in Ex Machina

-> based on this very limited selection of films, this claim is highly dubitable. Arguably, the transmission of affect has been part of cyborg imaginaries since Frankenstein and virtually every major science fiction narrative about cyborgs after.

383

…they ARE discussing…

384-386

...scrutinizing the proprieties of“this extreme, wholly other posthuman, though a common fixture in science fiction, remains at the periphery of texts which center on a hybrid posthuman who retains a very familiar “natural self” and is an extension of rather than “successor” to the human being”… 

-> double-check use of quotation marks in quote (single vs. double)

-> integration of quote is ungrammatical (requires a “which”)

391

merging the scientific and the humane that will eventually transcend all corporeal boundaries. 

-> unclear dichotomy: scientific and humane. 

-> alternatively: technological/human

-> also this dichotomy seems to perpetuate the very essentialist distinctions between the technological and the human, nature/culture, that a “posthuman approach” may want to leave behind.

Author Response

Dear Reviewer,

Thank you so much for your detailed review! Due to the extensiveness of revisions needed, I have rewritten my paper to focus on cross-examination of affect theory, posthumanism, and effective embodiment. I also took care to intersect textual analysis with theoretical discourse to allow a smooth transition between the two. I have revised my grammar usage and works cited to the best of my capabilities. Please do not hesitate to offer more thoughts/ opinions, and I look forward to your response concerning my revision.

Best Regards,

Celine Fahn

Reviewer 2 Report

The writer sets up an opposition between the human and the material (p.2 l.46) that could be done in a more nuanced fashion. Humanity has traditionally been seen as a site for the opposition of the 'human' (i.e., the mind) and the material (i.e. the body) rather than as a unity in opposition to materialism.

The writer might also like to review their assertion that 'much like the way computers processes data in command and response' (p.2) - there are many problems with comparing the human mind to a computer and showing some awareness of these issues is particularly important in conversations relevant to twenty-first century conceptions of the mind-body relationship, and the possibility of AI. This has implications for the development of the author's argument - on page 3 they describe the human body as 'an organic machine' - this should perhaps be revised to reflect that the author is talking about the TREATMENT of the human body as an organic machine. There is also no current example of a machine 'built to initiate organic thought', and this certainly isn't argued or evidenced in the paper up until this point.

The paragraph on the genome on page 2 was not particularly relevant to the paper in my opinion, but if it is to be kept it could perhaps reference the science fictional impact of the genome, e.g. Lars Schmeink's book on biopunk.

I would steer the author away from such claims as 'posthumanists believe' (5) and suggest that she rather uses 'Haraway argues' or similar.

I don't currently find the alignment of affect theory with geno-determinism on page 5 convincing. Affect theorists tend to talk about the circulation of affect (and sometimes emotions) as difficult to articulate, they do not generally speak about hormonal exchanges or of affects as specific material phenomenon. Apologies to the author if she knows of theorists who do this, but if so this needs to be more clearly articulated and more convincingly argued. Once again, the comparison of the body with a machine needs to be viewed with some critical distance.

During the reading of the film the author should consider looking at mise en scene and other cinematic techniques when discussing the film, particularly since the topic is affect. The current focus on dialogue and reading text from the script overlooks the affective techniques used by the film on the viewer, affective techniques that may open a more interesting reading than restricting the reading to Caleb's affective relationship with Eva, which is the current focus of the reading.

Further development of the conclusion is necessary - the significance of affective transmission to the writer's argument should be made clear on pages 9 and 10. The discussion of posthumanism here tends to generalise and does not add much to the author's argument up until this point. Highlighting the importance of affective transmission will help to strengthen the article's trajectory.

Author Response

Dear Reviewer,

Thank you so much for your detailed review! Due to the extensiveness of revisions needed, I have rewritten my paper to focus on cross-examination of affect theory, posthumanism, and effective embodiment. I also took care to intersect textual analysis with theoretical discourse to allow a smooth transition between the two. I have added paragraphs on affective scenes in the film as per request. I have revised my grammar usage and works cited to the best of my capabilities. Please do not hesitate to offer more thoughts/ opinions, and I look forward to your response concerning my revision.

Best Regards,

Celine Fahn

Round 2

Reviewer 1 Report

This is a great revision. The author has gone to great length to introduce relevant theoretical frameworks and integrate them with the discussion of the film. I have no hesitations about recommending the publication of this revised piece.

minor copy-edits:

11 initiate

28 “has” or cut “the role of”

69 in-depth

112 a disconnection

Author Response

Dear Reviewer,

Thank you so much for your kind remarks. Your advice in the first revision round gave me the inspiration and direction I needed, my revision would not have gone as smoothly without your help. I have completed the copy-edits to this paper as per your suggestion. The edits are highlighted in red for easy reference. Thank you again for your insight and guidance!

Sincere Regards,

C.W. Fahn

Reviewer 2 Report

The work put in to improving this essay in a short time is very impressive. The difficult concepts the paper deals with - affect theory, posthumanism, and artificial intelligence - are now clear and represented with nuance. They are also integrated with the reading of Ex Machina, producing a far stronger paper of greater originality and significance. My congratulations to the author. I spotted a few tiny things that could be changed, and the paper will benefit from copy-editing, but other than these tiny quibbles I recommend the paper for publication in its current form.

Descriptions of action in the film should be in present tense (e.g. on page 6, 'this changed very quickly' should be 'this changes').

Page 2 - I think the Herbrechter quote should mention 'interdisciplinarity' rather than 'interdisciplinary'

Page 3 - 'agues' should be 'argues'

Author Response

(The authors gave the same response as above.)
